# Comparative Analysis of Perceived Threat Threshold from Various Drivers to Cranes Along Indus Flyway, Punjab, Pakistan

**DOI:** 10.3390/biology14091275

**Published:** 2025-09-16

**Authors:** Ayesha Zulfiqar, Xueying Sun, Qingming Wu, Abdul Rehman, Nasrullah Khan, Mah Noor Khan

**Affiliations:** 1College of Wildlife and Protected Area, Northeast Forestry University, No. 26, Hexing Road, Xiangfang District, Harbin 150040, China or ayeshazulfiqar@uoj.edu.pk (A.Z.); xueyingsun@nefu.edu.cn (X.S.); 2Department of Zoology, University of Jhang, Jhang 35200, Pakistan; rehmanqurashi67@gmail.com (A.R.); mahnoorkhan@uoj.edu.pk (M.N.K.); 3College of Statistical Sciences, University of the Punjab, New Campus, Lahore 54590, Pakistan; nasrullah.stat@pu.edu.pk

**Keywords:** electrocution, human-subsidized predation, migratory cranes, poaching (illegal killing), poaching

## Abstract

The study evaluates the perceived anthropogenic threats to migratory crane species (*Grus grus* and *Grus virgo*) along the Indus Flyway in central and southern Pakistan, using 400 structured questionnaires administered to various stakeholders, including local wildlife officials, farmers, fishermen, hunters, shepherds, electrical technicians, students, boat skippers, shopkeepers, and local residents. The results revealed that poaching (illegal killing) emerged as the most frequent and severe perceived threat, with southern regions like Rajanpur facing the highest risks. Riverbanks particularly were identified as critical hotspots for poaching activities. The study also found that threat type and poaching method strongly influenced people’s perceptions, while factors like education, occupation, or crane species preference had a non-significant impact. The findings call for focused conservation efforts in high-risk regions, stronger enforcement against poaching, and community education to protect crane populations.

## 1. Introduction

The global human population surpassed 8 billion in 2022, intensifying pressure on ecosystems and accelerating biodiversity loss [1]. Vertebrate populations have declined by 68% since 1970, with 28% of assessed species now threatened with extinction [2,3,4]. Migratory birds face heightened risks during their annual cycles as urbanization and habitat fragmentation disrupt critical stopover sites [5,6]. Within this context, Pakistan’s Indus Flyway exemplifies conservation challenges as a vital corridor for over 400 migratory bird species [7,8].

Among these migratory species, cranes are particularly vulnerable due to their ecological sensitivity and cultural significance. Globally, 11 of 15 crane species face extinction risks [9]. While Pakistan historically hosted four crane species, only two now regularly utilize the Indus Flyway, the Demoiselle Crane (*Grus virgo*) and Eurasian Crane (*Grus grus*), maintaining an ecological presence in this corridor [10,11,12]. This change reflects the local extinction of the critically endangered Siberian Crane (*Leucogeranus leucogeranus),* with no validated records since 1999 [8,9,13], and the near extirpation of the resident Sarus Crane (*Antigone antigone*), now confined to isolated Sindh wetlands > 500 km south of our study area with <10 individuals nationally [14,15,16]. Consequently, despite their “Least Concern” status, both species (*Grus virgo* and *Grus grus*) face intensifying anthropogenic threats that jeopardize their regional persistence throughout the migratory corridor [10,11,17,18].

Immediate threats include poaching (illegal killing and live capture for illegal wildlife trade or taming), habitat loss, human-subsidized predation (e.g., stray dogs), and collision with power lines [11,12,19]. Climate change presents additional future risks [20]. The USD 12 billion annual illegal wildlife trade exacerbates these pressures, with cranes frequently targeted [19,21,22]. Despite legal protections, crane poaching persists across the country (Pakistan) due to subsistence needs and cultural practices [18,23]. These threats align with the EU Habitats Directive’s definitions of pressures (ongoing impacts) and threats (anticipated impacts) (EU 92/43/EEC).

Critical knowledge gaps hinder evidence-based conservation of crane species along the Indus Flyway [24]. While existing studies document poaching and habitat degradation, few quantify how socio-cultural drivers or spatiotemporal dynamics modulate threat severity. For instance, linkages between community education levels and poaching methods have been established for other avian taxa [25], the impact of socio-cultural and demographic factors on threats to crane species in Pakistan remains largely unexplored. The primary aim of this study is to assess the perceived anthropogenic threats to crane populations, including poaching, human-subsidized predation, and power line collisions in central and southern Pakistan. Additionally, the study explores how education levels, threat types, poaching methods, and occupation influence threat perceptions, rather than focusing solely on species, as most crane species, including *Grus virgo* and *Grus grus*, face similar threats across their range [12,26]. Though citizen science could address these gaps by engaging local stakeholders, such approaches remain underutilized in the region [27,28].

This study addresses these deficiencies through an integrated assessment of crane conservation challenges. We (1) evaluate and categorize the anthropogenic threats to crane populations based on their perceived frequency and severity; (2) identify high-risk zones based on citizen-reported data to guide targeted interventions; and (3) assess the impact of various factors, such as education level, occupation, threat type, and poaching methods, on public perceptions of the frequency and severity of these threats. We test the three following hypotheses: (i) Poaching (illegal killing) is likely to have a higher perceived frequency and severity compared to other anthropogenic threats; (ii) citizen-reported data will reveal threat hotspots overlooked by traditional monitoring; (iii) threat type and poaching method significantly influence the perception of threat frequency and severity. By integrating these objectives, we establish a replicable model for targeted conservation. This approach directly addresses identified gaps in socio-ecological threat assessment while balancing species protection with livelihood needs. Our findings inform adaptive policy under Punjab’s Wildlife Protection Act (2023) and provide transferable protocols for migratory corridors facing similar human–wildlife complexities.

## 2. Materials and Methods

### 2.1. Study Area

Pakistan features diverse landscapes, from mountain ranges to fertile plains, and supports extensive riverine ecosystems critical for resident and migratory bird species [24,29,30,31]. This study focuses on a semi-arid region of central and southern Pakistan encompassing eight districts, Attock, Mianwali, Khushab, Bhakkar, Layyah, Muzaffargarh, Dera Ghazi Khan, and Rajanpur, located along the Indus and Jhelum Rivers, which form part of the Indus Flyway (Figure 1). The area spans approximately 66,141 km^2^ between 29.10° N and 33.76° N latitudes and 70.32° E and 72.36° E longitudes, with altitudes ranging from 150 to 500 m. It experiences mean annual temperatures of 15 °C to 30 °C and monsoon-driven rainfall averaging 250–500 mm [32]. The region’s climate and riverine habitats provide essential stopover and foraging sites for migratory cranes. A total of 23 GPS-marked crane distribution points were selected as sampling sites/field sites, covering around 10,000 km^2^. Site selection was guided by field surveys and consultations with Wildlife Department staff and local communities, varying across districts based on crane presence.

### 2.2. Study Design

This study employed a multi-year, spatially explicit design to quantify local stakeholder perceptions of anthropogenic threats to migratory crane populations (*Grus virgo* and *Grus grus*) along the Indus Flyway. Fieldwork spanned six migratory seasons (Fall 2021–Spring 2024) and utilized a structured questionnaire approach (Appendix A). Given the scarcity of direct ecological monitoring data in the region, stakeholder perceptions were used as validated proxies for threat assessment, consistent with established frameworks for conservation prioritization in data-limited contexts [33,34,35,36,37]. This approach leveraged the unique strength of Local Ecological Knowledge (LEK) to detect cryptic threats such as clandestine poaching, which evade formal monitoring systems [38,39,40]. The methodology specifically focused on capturing perceived threat frequency and severity as community-validated indicators of pressure patterns requiring intervention, rather than absolute population impacts [41,42].

### 2.3. Sampling Strategy

A purposive sampling strategy was employed to target key stakeholder groups whose livelihoods, occupations, or proximity brought them into direct contact with crane habitats and associated pressures. These stakeholders included local residents, farmers, fishermen, shepherds, shopkeepers, hunters, wildlife officials, electrical technicians, boat skippers, and students. These groups possess critical ground-level insights, often inaccessible through formal monitoring, and their daily exposure to crane habitats facilitates the detection of emerging threats and the validation of pressure patterns across the landscape. Local residents (n = 77), farmers (n = 42), fishermen (n = 60), shepherds (n = 23), boat skippers (n = 37), and shopkeepers (n = 37) provided observations on threats like poaching and human-subsidized predation due to their daily exposure. Hunters (n = 65) offered crucial insights into poaching methods and spatial patterns despite the sensitivity of this information. Wildlife officials (n = 30) and students (n = 29) provided valuable information based on their regulatory experience, local knowledge, and general understanding of crane threats within their region, while electrical technicians (n = 18) provided data on power line collision-related mortality. This multi-stakeholder approach intentionally integrated diverse knowledge systems from experiential LEK to technical expertise to ensure a comprehensive understanding of threats vital for conservation planning [43,44,45].

The total sample size of 400 respondents (used for data analysis) was determined using a multi-factorial approach. Primarily, it ensured robust spatial coverage for analyzing threat heterogeneity across eight districts, with a target of at least 50 respondents per district. This district-level sample size was estimated to provide adequate statistical power (>0.8) for detecting significant differences (α = 0.05) in perceived threat levels between districts, based on variance estimates from pilot surveys and established thresholds for socio-ecological research. Secondarily, participant allocation within districts captured stakeholder-specific perspectives while reflecting their relative prevalence, with minimum group sizes (e.g., n = 18 for electricity workers) set to enable meaningful analysis. The overall sample aligns with conservation social science precedents, where multi-stakeholder assessments engage 200–600 respondents to achieve thematic saturation within geographic/stakeholder strata while enabling robust statistical comparisons [46,47]. Selection within districts prioritized proximity to GPS-marked crane sites (n = 23 sites across ~10,000 km^2^), contextual ecological knowledge validated by local community leaders or Wildlife Department staff, and recommendations to identify knowledgeable participants. Reflecting regional socio-cultural norms, participants were predominantly male due to their primary roles in wildlife-associated outdoor activities.

To ensure that respondents could accurately identify crane species, visual identification guides were systematically employed during in-person interviews. Prior to answering threat assessment questions, respondents demonstrated familiarity by correctly identifying *Grus grus* (Eurasian Crane) and *Grus virgo* (Demoiselle Crane) using standardized images, describing key distinguishing features (e.g., size, plumage), and pronouncing the local names “Bari Koonj” and “Choti Koonj”, commonly used for both species. Only data from respondents confirming basic species recognition were included in analyses requiring taxon-specific attribution. Given the significant overlap in migratory routes and shared vulnerability to common threats, the core analyses aggregated data from both crane species, unless examining species-specific patterns [48].

Data collection was guided by socio-ecological systems theory focusing on the anthropogenic pressures [49]. Key threats including poaching, electrocution, and predation were conceptually defined as observable human-driven pressures reported to cause crane mortality [11,19,20]. These threats were operationalized using standardized protocols adapted from the IUCN Threat Classification Scheme and validated crane studies [50]. Given the absence of formal records on poaching and other threats (as these are often clandestine and not systematically documented), the study relied entirely on questionnaire data. Threat frequency was measured via a 5-point Likert scale (1 = “Very Rare” to 5 = “Very Frequent”) asking, “How often do you observe the threats affecting cranes here during migration seasons?”, while severity used a parallel scale (1 = “Not Severe” to 5 = “Extremely Severe”) probing perceived impacts, with both scales derived from established conservation protocols.

Socio-cultural drivers included education level using Pakistan Census categories [32], livelihood dependence proxies, occupation, and hunting methods coded via regional ethno-zoological classifications. Sensitive topics, particularly poaching, were approached carefully to minimize discomfort and avoid eliciting self-incriminating responses. Rather than inquiring about personal involvement, questions focused on general community-level observations (e.g., “What poaching methods are commonly used?”). Follow-up questions explored motivations, including subsistence needs and cultural beliefs. All procedures adhered to strict ethical standards approved by the Institutional Research Ethics Committee (Ref No./UOJ/Ethical Committee/02), with verbal informed consent, confidentiality assurances, voluntary participation, and anonymity through non-collection of identifiers. This ethical grounding aligned with the Declaration of Helsinki and facilitated trust necessary for obtaining reliable data on illicit activities.

The reliance on perceptual data is explicitly justified as a critical proxy where direct ecological monitoring is limited [51]. Local Ecological Knowledge (LEK) systematically captures community-validated pressure patterns, detecting cryptic threats that evade formal surveillance [52]. While LEK provides invaluable insights into threat patterns, it is emphasized that these findings reflect stakeholder perceptions rather than absolute measures of population impact. These perceptions serve as important indicators of conservation priorities requiring intervention, with triangulation further strengthening ecological validity where feasible. This participatory approach integrates diverse knowledge systems, providing a robust foundation for identifying conservation priorities across the Indus Flyway.

### 2.4. Statistical Analysis

All statistical analyses were conducted using R software (version 4.3.1; R Core Team 2023) [53], beginning with rigorous data preprocessing that excluded seventeen of the 417 initial responses due to missing values or logical inconsistencies such as high severity ratings paired with “never observed” frequency designations. Categorical variables including region (8 districts), threat type (5 categories), and hunting method (4 types) were factorized, with Poaching established as the threat baseline and Rajanpur as the regional reference. For continuous variables, Likert-scale responses measuring perceived frequency and severity (1–5 scale) were analyzed as ordinal continuous variables, while Hunting Parties counts underwent log-transformation [log(x + 1)] to correct right-skewness, evidenced by Shapiro–Wilk normality test improvements from W = 0.92 pre-transformation to W = 0.98 post-transformation. Threat hierarchy assessment employed Kruskal–Wallis tests, confirming significant overall differences in both threat frequency (χ^2^ = 210.5, *p* < 0.001) and severity (χ^2^ = 225.8, *p* < 0.001). Due to violated normality assumptions (Shapiro–Wilk *p* < 0.001), post hoc pairwise comparisons were conducted using Dunn’s procedure (1964), with *p*-values adjusted via the Bonferroni method, while effect sizes quantified through Cohen’s d revealed Poaching (illegal killing) exhibited the largest discrepancy from human-subsidized predation (frequency: d = 2.62; severity: d = 2.93). Regional variations were evaluated through one-way ANOVA after verifying critical assumptions: Levene’s test confirmed variance homogeneity (*p* = 0.21 for frequency; *p* = 0.18 for severity), and Durbin–Watson statistics (1.93) indicated residual independence, with significant regional effects for frequency (F(7, 392) = 6.326, *p* < 0.001) and severity (F(7, 392) = 3.204, *p* = 0.003) prompting additional Dunn’s post hoc tests showing maximal differences between Rajanpur and Khushab (frequency Δ = 1.12, adjusted *p* < 0.001; d > 0.8). Spearman’s rank-order correlations with bootstrapped 95% CIs (1000 iterations) assessed frequency–severity relationships per threat type, revealing near-perfect alignment for collision with power lines (ρ = 0.93 [0.84, 0.97]). For spatial clustering, K-means incorporated z-standardized threat frequency, severity, and poaching party counts, with optimal k = 3 determined through the elbow method (within-cluster sum of squares reduction < 5% beyond k = 3), silhouette width > 0.5, and Calinski–Harabasz index (210.7) confirming separation, while cluster stability was validated via 1000 bootstrap replicates (Jaccard similarity > 0.85). Two-way ANOVA models evaluating perception determinants demonstrated threat type dominance for both frequency (F = 104.92, *p* < 0.001, partial η^2^ = 0.62) and severity (F = 153.64, *p* < 0.001, partial η^2^ = 0.71), with model fitness verified through residual diagnostics (Q-Q plots confirming normality), variance inflation factors < 1.8 (no multicollinearity), adjusted R^2^ values (0.84 for severity models), and AIC comparisons against reduced models (ΔAIC > 10 favoring full models). Post hoc comparisons were adjusted using the Benjamini–Hochberg false discovery rate method. Sensitivity analyses employing non-parametric alternatives yielded consistent results, confirming the robustness of findings. Post hoc power analyses, conducted in G*Power 3.1 [54], indicated statistical power exceeded 95% for all ANOVA models (α = 0.05, f = 0.25).

## 3. Results

### 3.1. Poaching (Illegal Killing) Ranked Highest in Perceived Threat Hierarchy: Frequency and Severity Analysis of Anthropogenic Risks to Crane Populations

The study was carried out to assess the perceived anthropogenic threats to cranes in central and southern Pakistan by surveying 400 local participants through structured questionnaires (Appendix A). The respondents’ ages ranged from 18 to 70 years, with a median age of 39, ensuring a broad representation of community perspectives. The findings highlight key threats based on their perceived frequency and severity, providing critical insights for conservation strategies.

The data reveals that poaching (illegal killing) is perceived as the most severe and frequent threat (frequency = 4.9, severity = 4.8), followed closely by illegal wildlife trading (frequency = 4.7, severity = 4.5). Taming practices show moderately high concern (frequency = 4.6, severity = 4.3), while collision with power lines presents intermediate threat levels (frequency = 4.4, severity = 4.0). In contrast, human-subsidized predation consistently scores lowest (frequency = 2.3, severity = 2.2), reflecting its minimal perceived impact on crane populations (Table 1).

The post hoc analysis confirms poaching (illegal killing) as the most significant threat, showing strong statistical differences in both frequency and severity (all *p* < 0.001). Frequency perceptions establish clear patterns, with poaching (illegal killing) demonstrating the largest difference compared to human-subsidized predation (diff = 2.62, *p* < 0.001), followed by its substantial advantage over collision with power lines (diff = 1.80, *p* < 0.001). While illegal wildlife trade maintains clear frequency distinctions from human-subsidized predation (diff = 1.48, *p* < 0.001), it shows no significant difference from taming (diff = −0.04, *p* = 0.999). Human-subsidized predation consistently emerges as the least frequent threat, particularly when compared to collision with power lines (diff = −0.82, *p* = 0.009) (Table 2).

The severity analysis mirrors and intensifies these patterns, with poaching (illegal killing) showing even greater dominance (all *p* < 0.001). The severity gap between poaching (illegal killing) and human-subsidized predation (diff = 2.93) represents the largest measured difference, followed by poaching (illegal killing) advantage over power line collision (diff = 1.94). Illegal wildlife trade maintains its intermediate position—significantly more severe than human-subsidized predation (diff = 1.48) but statistically indistinguishable from taming (diff = 0.23, *p* = 0.506). The consistent negative values for human-subsidized predation comparisons (particularly diff = -0.98 against collision) quantitatively confirm its status as the least concerning threat across both dimensions (Table 3).

Building upon the established threat hierarchy, correlation analysis reveals strong relationships between perceived frequency and severity for most threats (Table 3). Collision with power lines demonstrates the strongest correlation (r = 0.930, *p* < 0.001), indicating near-perfect alignment between how frequently it is observed and how severely it is perceived, followed closely by illegal wildlife trading (r = 0.915) and taming (r = 0.868). Poaching (illegal killing) shows a moderate but still highly significant correlation (r = 0.557, *p* < 0.001), suggesting that while universally recognized as the most severe threat, its perceived frequency varies more independently of severity judgments. Human-subsidized predation stands apart with no significant correlation (r = 0.189, *p* = 0.482), confirming its unique status as a threat where occurrence and perceived impact show no consistent relationship (Table 4).

### 3.2. Identification of Regional High-Risk Zones for Prioritized Intervention

ANOVA results indicated significant regional variations in both the frequency (F(7, 392) = 6.326, *p* < 0.001) and severity (F(7, 392) = 3.204, *p* = 0.0026) of perceived threats to crane populations, confirming an uneven distribution across the surveyed regions. Rajanpur emerged as a significantly affected region, reporting the highest mean scores for both frequency (4.88) and severity (4.82). Conversely, Khushab recorded the lowest scores for both threat dimensions (frequency: 3.76; severity: 4.02). Mianwali ranked second-highest after Rajanpur in both frequency (4.68) and severity (4.48). Regions including Attock, DG Khan, Muzaffargarh, Layyah, and Bhakkar demonstrated intermediate levels of perceived threat, as detailed in Table 5.

To identify specific pair-wise differences, Dunn’s post hoc test with Bonferroni correction was applied. This analysis revealed that Khushab exhibited significantly lower perceived threat frequency compared to Attock, DG Khan, Layyah, Mianwali, Muzaffargarh, and Rajanpur, with the most pronounced contrast observed between Khushab and Rajanpur (*p* = 1.22 × 10^−8^). Moreover, Bhakkar reported significantly lower frequency than Rajanpur (*p* = 9.22 × 10^−3^), further reinforcing Rajanpur’s status as the most impacted region. Regarding severity, significant differences emerged specifically between Rajanpur and Bhakkar (*p* = 0.0155) and Rajanpur and Khushab (*p* = 0.00098), indicating that Rajanpur experiences not only more frequent but also more severe perceived threats. No further significant differences in severity were detected among the other regions, suggesting comparable levels elsewhere. The pronounced intensity of perceived threats in Rajanpur likely reflects localized risk factors warranting targeted investigation. Collectively, these post hoc results substantiate the ANOVA findings, confirming notable spatial disparities where Khushab consistently reports the lowest perceived threat levels, and Rajanpur emerges as the region of greatest vulnerability. Comprehensive Dunn’s post hoc test results are detailed in Appendix A.

#### 3.2.1. Spatial Distribution of Perceived Threats (Location-Based Analysis)

Further analysis pinpointed the river bank as the primary geographical locus for perceived threats to cranes. At this critical habitat, poaching (illegal killing) constituted the most significant threat, with 42 recorded instances, underscoring its predominance as a regional risk factor. Supplementary anthropogenic pressures included captive keeping (“Taming”; three instances) and illegal wildlife trading (“Trading”; four instances), collectively highlighting substantial human-driven interactions impacting crane populations. Additional observed threats comprised human-subsidized predation (one instance) and collision with power lines (one instance) (Table 6).

#### 3.2.2. Poaching Camps and Their Role in Perceived Threat Intensification

Poaching camps intensified threats spatially, with notable disparities in activity levels. Camps 4, 5, 6, and 7 functioned as high-intensity hubs, averaging two to nine poaching parties, thereby concentrating organized poaching pressure. Conversely, Camps 2 and 3 demonstrated markedly lower activity. A minority of respondents reported no camp affiliation (“None”; average: two parties), indicating opportunistic independent poaching (Table 7). Critically, stakeholders identified these camps, particularly high-activity hubs (4–7), as primary catalysts for regional poaching prevalence, directly linking organized camp operations to intensified poaching (Table 7).

#### 3.2.3. Identification and Characterization of Perceived Threat Categories by Cluster Analysis

In order to further dissect the perceived threats to crane populations and gain a deeper understanding of their distribution and intensity, K-Means clustering analysis was conducted. This method grouped the data based on three key variables: threat frequency, severity, and the involvement of poaching camps (represented by poaching parties). The analysis revealed three distinct threat clusters: Cluster 1 (32 instances), Cluster 2 (5 instances), and Cluster 3 (13 instances). These clusters were characterized by specific patterns in threat frequency and severity, providing a clearer picture of regional poaching risks.

Cluster 1, which contained the largest number of instances (32), displayed the highest frequency and severity of perceived threats. These clusters are linked to moderate poaching party involvement, highlighting that poaching activities here are frequent and severe. In stark contrast, Cluster 2 (four instances) exhibited extremely low levels of both frequency and severity. The negative poaching party involvement (essentially a lack of organized poaching activity) suggests that the perceived threats in this cluster are relatively minor and not immediately critical. However, the low threat levels do not imply that these areas should be ignored. Continuous monitoring of these regions is still recommended to detect any potential shifts in poaching patterns over time. This proactive approach could prevent the escalation of threats that may arise unexpectedly.

Cluster 3, which consisted of 13 instances, displayed moderate levels of threat frequency and severity. It also involved moderate poaching activity, suggesting that these areas require a balanced mitigation approach. While the threat levels are not as alarming as those in Cluster 1, they still warrant close monitoring and targeted intervention to prevent potential escalation into more severe threats, as seen in Cluster 1.

The clustering structure was further evaluated through Within-Cluster Sum of Squares (WSS). The WSS values for each cluster were as follows: 7.43 for Cluster 1, 17.49 for Cluster 2, and 8.38 for Cluster 3. These values suggest that Cluster 1 and Cluster 3 exhibit lower within-cluster variability, while Cluster 2 has higher variability, reflecting the nature of its low threat levels.

The between-cluster variance accounted for 77.3% of the total variation, indicating that the three clusters are distinctly separated from each other. To ensure that the number of clusters chosen was optimal, the elbow method was applied, which revealed that three clusters provided the best balance between within-cluster compactness and between-cluster separation (Figure 2). Beyond three clusters, the reduction in WSS was marginal, confirming that k = 3 was indeed the most appropriate choice.

Figure 3 visually represents the spatial distribution of the clusters based on perceived threat frequency and severity. The clusters are color-coded for clarity: Cluster 1 (Blue) represents areas with high-frequency and high-severity threats, Cluster 2 (Pink) illustrates areas with low-frequency and low-severity threats, and Cluster 3 (Yellow) depicts areas with moderate threat levels. This spatial differentiation further highlights the need for region-specific conservation strategies.

### 3.3. Analysis of Threat Perception Determinants in Crane Conservation

#### 3.3.1. Perceived Frequency of Threats: Significant/Non-Significant Determinants

The statistical analysis reveals clear patterns in how different factors influence threat frequency perceptions among respondents. Threat type stands out as the most powerful determinant, demonstrating an exceptionally strong effect (F = 104.92, *p* < 0.001) that accounts for the majority of explained variance (222.725 Sum Sq). This indicates clear public differentiation between threat frequencies, particularly recognizing poaching (illegal killing) as more frequent than human-subsidized predation. Poaching methods also significantly affected perceptions (F = 10.14, *p* < 0.001), suggesting specific techniques like Shooting influence frequency judgments shown in Table 8.

In contrast to these strong ecological factors, demographic characteristics show minimal influence on threat frequency perceptions. While occupation approached marginal significance (F = 1.64, *p* = 0.103), education level (F = 0.57) and species preference (F = 0.38) proved irrelevant (both *p* > 0.05). These results demonstrate that threat frequency perceptions are shaped by ecological realities rather than respondent characteristics.

#### 3.3.2. Perceived Severity of Threats: Significant/Non-Significant Determinants

The analysis of severity perceptions reveals even stronger patterns than those observed for frequency. Threat type demonstrated significant influence on severity perceptions (F = 153.64, *p* < 0.001), accounting for 270.52 units of variance. Poaching method also showed significant effects (F = 15.43, *p* < 0.001) with 20.38 Sum Sq. Demographic variables were non-significant: Occupation (F = 1.04, *p* = 0.407), Education Level (F = 0.43, *p* = 0.789), and Preferred Crane Species (F = 1.17, *p* = 0.310) (Table 9).

Citizen science results demonstrate that community-reported threats reliably identify poaching (illegal killing) as the most severe and frequent risk to cranes (frequency = 4.9, severity = 4.8). The strong alignment between local perceptions and spatial clustering patterns (77.3% variance explained) validates participatory monitoring as an effective tool for prioritizing anti-poaching interventions. This approach offers a practical, community-driven framework for targeted crane conservation.

## 4. Discussion

### 4.1. Integrated Threat Hierarchy and Spatial Prioritization

Our study quantifies anthropogenic threats to migratory cranes along Pakistan’s Indus Flyway, revealing poaching (illegal killing) as the dominant threat in both frequency (M = 4.9) and severity (M = 4.8), with statistically extreme differences from all other threats (pairwise *p* < 0.001; Cohen’s d > 2.6 vs. human-subsidized predation). This aligns with global crane decline patterns [55,56,57,58] and regional catastrophes like South Waziristan’s mass captures [17,18,23]. The consistency of this hierarchy across 400 respondents spanning ages 18–70 (median 39) underscores poaching’s (illegal killing) entrenched role in driving crane mortality.

### 4.2. Illicit Trade and Taming: Underestimated Localized Threats

Illegal wildlife trading (frequency: 4.7; severity: 4.5) and taming (4.6/4.3) form a secondary threat tier. Though statistically indistinguishable in severity (*p* = 0.506), both exceed power line collisions (4.4/4.0) and human-subsidized predation (2.3/2.2). Driven by cultural demand [58,59] and commercial breeding [11,23], these practices thrive where enforcement is weak (e.g., Bhakkar, Mianwali). Their moderate correlation between frequency and severity (trade: r = 0.915; taming: r = 0.868; Table 4) suggests communities accurately gauge their impacts. Globally, unregulated wildlife trade accelerates species decline, emphasizing the need for policy interventions to curb demand and disrupt trafficking networks [60,61].

### 4.3. Poaching Networks: Scale and Sophistication

Organized poaching camps (Table 7) intensify threats spatially, with Camps 4–7 acting as high-intensity hubs (6.7–9 parties on average). This systematization explains poaching dominance in threat hierarchies and correlates with regional mortality hotspots. The spatial concentration of camps near riverbanks (42/50 reported poaching incidents; Table 6) mirrors patterns in Khyber Pakhtunkhwa [62,63], where decoy systems [18,64,65] and riverine ambushes escalate efficiency. Such operational sophistication demands tactical enforcement beyond routine patrols [17,66].

### 4.4. Regional Disparities and Cluster-Based Vulnerability

Rajanpur emerged as the highest-risk district, with significantly elevated threat perception (frequency: 4.88; severity: 4.82; *p* < 0.001 versus Khushab and Bhakkar; Table 5) [67]. This spatial vulnerability was systematically validated through K-means clustering, which explained 77.3% of between-cluster variance and identified three distinct risk tiers: Cluster 1 (high-risk: 32 instances) exhibited peak threat levels linked to active poaching camps; Cluster 3 (moderate-risk: 13 instances) warranted preventive interventions; and Cluster 2 (low-risk: five instances) required ongoing monitoring. This analytical hierarchy confirms stakeholder perceptions and precisely delineates priority zones, with Rajanpur representing the epicenter of threat intensity, followed by secondary hotspots in Mianwali (frequency: 4.68; severity: 4.48) and critical river corridor habitats where poaching incidents occurred (Table 6).

### 4.5. Determinants of Threat Perception: Ecological over Demographic

Threat type (ANOVA frequency: F = 104.92; severity: F = 153.64; *p* < 0.001) and poaching methods (frequency: F = 10.14; severity: F = 15.43; *p* < 0.001) were the sole significant perception drivers (Table 8 and Table 9). Demographic factors (education, occupation, species preference) showed no statistical influence (*p* > 0.05), contradicting assumptions that literacy dictates conservation awareness. Instead, direct exposure to ecological threats not socio-educational status shapes community risk assessment [68].

### 4.6. Conservation Recommendations and Future Directions

Our findings dictate a four-pronged conservation strategy beginning with poaching-centric enforcement targeting high-activity camps (4–7) through intelligence-led operations under Punjab’s Wildlife Protection Acts, coupled with real-time monitoring of river corridors where Cluster 1 threats concentrate. Regionally tailored interventions must prioritize Rajanpur and Mianwali, combining enhanced patrols with community engagement to combat illegal wildlife trade and taming while replicating Khushab/Bhakkar’s compliance models through habitat incentives in lower-risk zones. Awareness initiatives should transcend demographic barriers by designing universal programs that highlight poaching’s ecological impact through culturally resonant channels (e.g., religious leaders), bypassing literacy limitations identified in our ANOVA results (Table 8 and Table 9). Finally, citizen science integration leverages strong threat–severity correlations (r = 0.557–0.930; Table 4) to establish participatory monitoring systems, expanding spatial coverage to adjoining flyways using cluster-based risk models (Figure 3) for proactive threat mitigation.

## 5. Conclusions

This study demonstrates that poaching (illegal killing) is the most severe and frequent threat to migratory cranes (*Grus grus* and *Grus virgo*) along Pakistan’s Indus Flyway, with significantly higher risk in southern regions, particularly Rajanpur district, where organized poaching camps near riverbanks drive localized population declines. Crucially, socio-demographic factors (e.g., education, occupation and crane species preference) do not shape threat perception; instead, threat type and poaching methods universally dictate stakeholder awareness. Based on this stakeholder-validated threat mapping, key conservation imperatives emerge. Targeted anti-poaching enforcement must be prioritized in high-risk districts like Rajanpur and Mianwali, specifically focusing on dismantling the identified four to seven high-activity poaching camps through intelligence-led operations empowered by Punjab’s Wildlife Protection department. Complementing enforcement, universal education programs are essential to reduce demand for crane taming and illegal trade; these programs should strategically bypass literacy barriers by utilizing religious and cultural channels, particularly targeting farmers, fishers, and hunters. Spatial management strategies should replicate the successful compliance model observed in Khushab district within lower-threat areas while simultaneously enhancing monitoring efforts in moderate-risk zones identified as Cluster 3. Furthermore, protecting critical river corridor habitats, which function as major poaching hotspots, requires implementation through real-time surveillance and community patrols. Collectively, these strategies provide a scalable framework for crane conservation across migratory corridors. Future conservation efforts must rigorously evaluate the cost-effectiveness of these interventions and their subsequent impacts on crane demography to ensure the long-term recovery of these populations.

## Figures and Tables

**Figure 1 biology-14-01275-f001:**
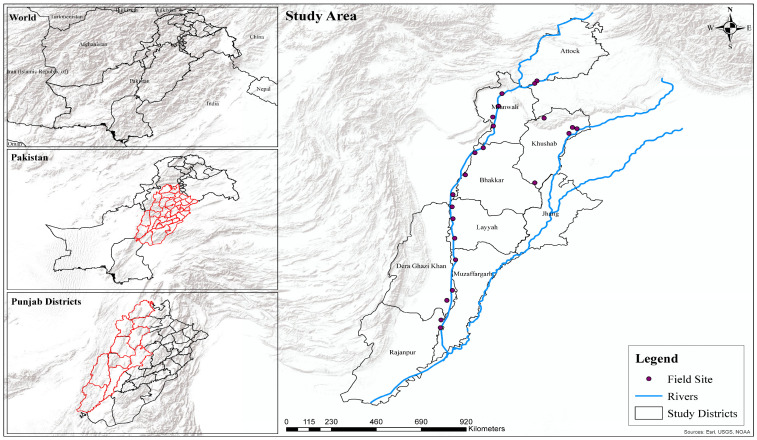
Location map of the study area along Indus Flyway, Pakistan.

**Figure 2 biology-14-01275-f002:**
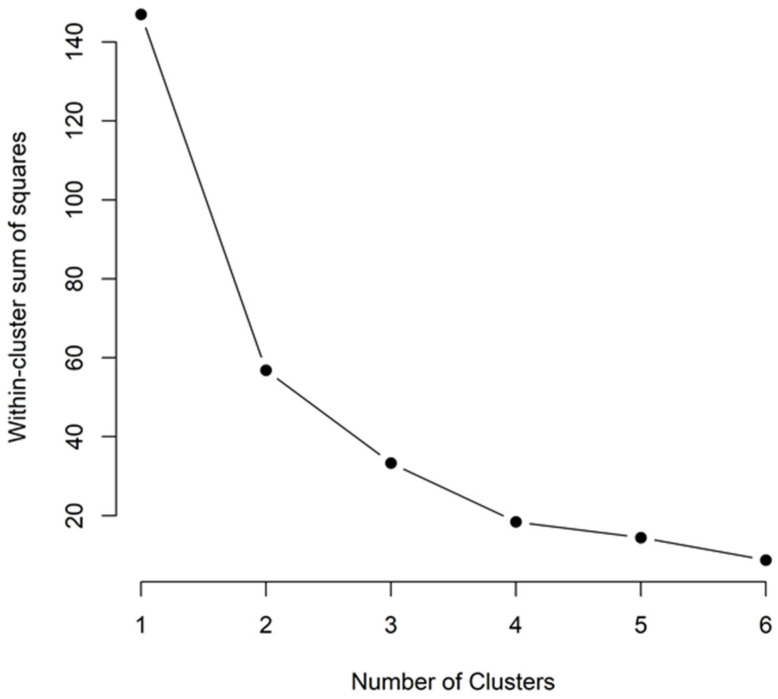
Elbow method plot illustrating the optimal number of clusters based on Within-Cluster Sum of Squares.

**Figure 3 biology-14-01275-f003:**
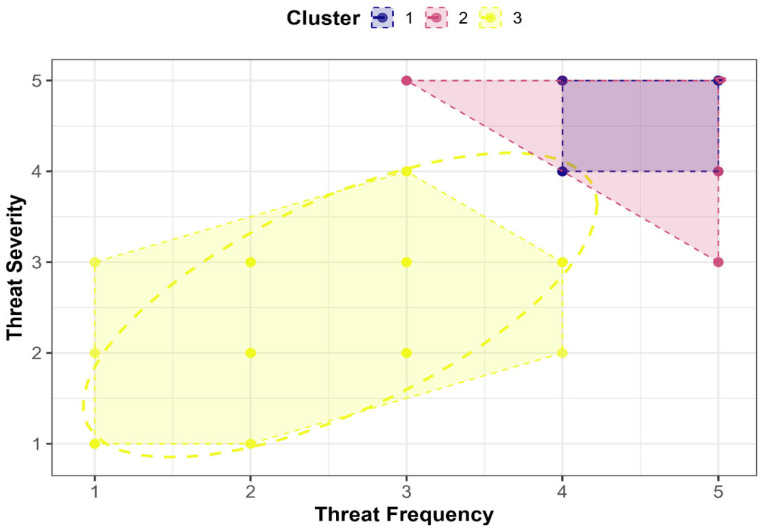
Plot showing the clustering of threat data based on threat frequency and severity.

**Table 1 biology-14-01275-t001:** Perceived threat levels to crane populations showing mean frequency and severity ratings for different anthropogenic threats (scale: 1 = lowest, 5 = highest).

Threat Type	Mean Frequency	Mean Severity
Poaching (Illegal Killing)	4.9	4.8
Illegal Wildlife Trading	4.7	4.5
Taming	4.6	4.3
Collision with Power Lines	4.4	4.0
Human-subsidized Predation	2.3	2.2

**Table 2 biology-14-01275-t002:** Pairwise comparisons of perceived threat frequency with 95% confidence intervals.

Threat Comparison	Mean Difference	Lower CI	Upper CI	*p*-Value
Poaching (illegal killing)—Human-subsidized predation	2.6200	2.0860	3.1541	<0.001
Poaching (illegal killing)—Collision with power lines	1.8021	1.3509	2.2533	<0.001
Illegal trade—Human-subsidized predation	1.4754	0.8444	2.1064	<0.001
Taming—Human-subsidized predation	1.4375	0.8678	2.0072	<0.001
Poaching (illegal killing)—Illegal wildlife trade	1.1447	0.7612	1.5281	<0.001
Illegal trade—Collision with power lines	0.6574	0.0948	1.2201	0.013
Taming—Collision with power lines	0.6196	0.1266	1.1125	0.006
Taming—Illegal wildlife trade	−0.0379	−0.4697	0.3939	0.999
Human-subsidized predation—Collision with power lines	−0.8179	−1.4922	−0.1436	0.009
Taming—Poaching (illegal killing)	−1.1825	−1.4536	−0.9115	<0.001

**Table 3 biology-14-01275-t003:** Pairwise comparisons of perceived threat severity with 95% confidence intervals.

Threat Comparison	Mean Difference (Diff)	Lower CI (Lwr)	Upper CI (Upr)	Adjusted *p*-Value (*p* Adj)
Poaching (illegal killing)—Human-subsidized predation	2.9256	2.4327	3.4185	<0.001
Poaching (illegal killing)—Collision with power lines	1.9446	1.5282	2.3610	<0.001
Taming—Human-subsidized predation	1.7138	1.1880	2.2397	<0.001
Illegal trade—Human-subsidized predation	1.4830	0.9006	2.0654	<0.001
Poaching (illegal killing)—Illegal wildlife trade	1.4426	1.0887	1.7966	<0.001
Taming—Collision with power lines	0.7328	0.2779	1.1878	<0.001
Illegal trade—Collision with power lines	0.5020	−0.0173	1.0213	0.064
Taming—Illegal wildlife trade	0.2309	−0.1677	0.6294	0.506
Human-subsidized predation—Collision with power lines	−0.9810	−1.6033	−0.3586	<0.001
Taming—Poaching (illegal killing)	−1.2118	−1.4620	−0.9616	<0.001

**Table 4 biology-14-01275-t004:** Correlation between perceived frequency and severity of anthropogenic threats.

Threat Type	Correlation (r)	*p*-Value	95% CI (Low)	95% CI (High)	Method
Collision with power lines	0.930	<0.001	0.8395	0.9702	Spearman
Illegal wildlife trading	0.915	<0.001	0.8333	0.9575	Spearman
Poaching (Illegal Killing)	0.557	<0.001	0.4652	0.6365	Spearman
Taming	0.868	<0.001	0.7991	0.9146	Spearman
Human-subsidized predation	0.189	0.482	−0.3380	0.6263	Spearman

**Table 5 biology-14-01275-t005:** Regional comparison of perceived threat frequency and severity to cranes, with Rajanpur showing the highest average values and lowest variability among surveyed districts.

Region	Mean Frequency	SD Frequency	Mean Severity	SD Severity	Count
Attock	4.62	1.01	4.62	1.01	50
Bhakkar	4.1	1.25	4.1	1.25	50
DG Khan	4.54	0.862	4.5	0.974	50
Khushab	3.76	1.15	4.02	1.22	50
Layyah	4.22	1.27	4.26	1.27	50
Mianwali	4.68	0.768	4.48	0.863	50
Muzaffargarh	4.44	1.05	4.48	1.01	50
Rajanpur	4.88	0.521	4.82	0.72	50

**Table 6 biology-14-01275-t006:** Distribution of perceived crane threats at primary geographical locus (river bank), highlighting poaching (illegal killing) as the predominant risk factor.

Location of Threat	Threat Type	Count
River Bank	Taming	3
River Bank	Poaching (illegal killing)	42
River Bank	Human-subsidized predation	1
River Bank	Illegal wildlife trading	3
River Bank	Collision with power lines	1

**Table 7 biology-14-01275-t007:** Average number of poaching parties observed across different crane poaching camps, with Camps 4 to 7 showing the highest activity levels.

Poaching Camps	Average Poaching Parties	Count
Camp 2	3.8	5
Camp 3	6	2
Camp 4	6.66	15
Camp 5	8	14
Camp 6	8.4	10
Camp 7	9	1
None	2	3

**Table 8 biology-14-01275-t008:** Two-way ANOVA results for perceived threat frequency across demographic and threat-related factors. Dependent variable: Perceived frequency of threats.

Source of Variation	df	Sum Sq	Mean Sq	F-Value	*p*-Value
Threat Type	4	222.73	55.68	104.92 ***	<0.001
Occupation	9	7.82	0.87	1.64	0.103
Education Level	4	1.21	0.30	0.57	0.685
Preferred Crane Species	2	0.40	0.20	0.38	0.687
Poaching Method	3	16.15	5.38	10.14 ***	<0.001
Residuals	377	200.08	0.53		

Note: Significance code: *** *p* < 0.001.

**Table 9 biology-14-01275-t009:** Two-way ANOVA results for perceived threat severity across demographic and threat-related factors. Dependent variable: Perceived severity of threats.

Source of Variation	df	Sum Sq	Mean Sq	F-Value	*p*-Value
Threat Type	4	270.52	67.63	153.64 ***	<0.001
Occupation	9	4.12	0.46	1.04	0.407
Education Level	4	0.75	0.19	0.43	0.789
Preferred Crane Species	2	1.03	0.52	1.17	0.310
Poaching Method	3	20.38	6.79	15.43 ***	<0.001
Residuals	377	165.95	0.44		

Note: Significance code: *** *p* < 0.001.

## Data Availability

The original contributions presented in this study are included in the article. Further inquiries can be directed to the corresponding author.

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
