# Peer review of "Comparative Analysis of Perceived Threat Threshold from Various Drivers to Cranes Along Indus Flyway, Punjab, Pakistan"

_biology, 2025, doi:10.3390/biology14091275_

Round 1
Reviewer 1 Report
Comments and Suggestions for Authors
Hi,
thanks for your efforts,
I suggest to use new content in Introduction and Methods sections about "citizen science" and "crowdsourcing". You should insist on the knowledge derived by citizens.
Plus, find few suggestions on the ms.
good luck,
M.K.

Author Response
Dear Reviewer,
Thank you very much for your time and constructive feedback on our manuscript titled "Comparative analysis of threats threshold from various drivers to Cranes along Indus Flyway, Punjab, Pakistan." We greatly appreciate your thoughtful comments, which have significantly contributed to improving the clarity and quality of the manuscript.
We have carefully addressed all the comments point by point. Please find attached the detailed response document, which outlines our responses to each comment, along with the corresponding revisions made in the manuscript. For your convenience, all changes in the manuscript are clearly highlighted by yellow highlighter and the page number and line number is clearly mention in response document.
We believe that the revisions have strengthened the manuscript, and we hope that it is now suitable for publication in the journal.
Thank you once again for your valuable suggestions and your continued support.

Reviewer 2 Report
Comments and Suggestions for Authors
General comments
This manuscript addresses an important topic concerning the threats faced by crane populations in Punjab, Pakistan, particularly focusing on poaching and other human-related pressures. The manuscript is based on 3 years of data, which is impressive for any monitoring study. However, the manuscript requires significant improvement in the objective definition to ensure clarity and more importantly, in the methods used to ensure scientific rigor and reproducibility.
The manuscript would benefit from a stronger, more coherent narrative throughout the introduction that clearly articulates the objective of describing temporal (i.e., frequency) and spatial (i.e., across districts) variation in poaching severity, as well as understanding the relative importance of social and cultural factors driving these patterns. The authors should also explicitly connect their objectives with previous research findings on the social drivers of poaching.
Across the method section, there is a lack of detail about the questionnaires and the stakeholder groups involved. It was not clear who exactly participated, nor how these groups were selected or represented. This makes it difficult to interpret the results and evaluate their broader applicability.
Also, key concepts such as “severity” and “frequency” of threats are not clearly defined or explained in terms of measurement. It remains unclear how these metrics were derived from the questionnaire data, or what specific questions were asked to obtain them. The manuscript currently lacks information on measures taken to ensure truthful reporting and worryingly, there is no mention of ethical approval for an external commitee to guarantee participant protection.
Finally, but not less important, the description of the analysis is far too superficial to allow replication. Important details about data preparation, coding of variables, model specifications, and model evaluation are missing (see details bellow).
Detailed comments
Summary
L11 – L12: Please, mention to who the questionaries were applied? Locals communities? NGO members? Researchers?
L13: Replace “danger” by “threat”
L15 – L16: Could please, rephrase this to make it more specific? “Education level and local involvement strongly influence both perception and practices related to crane threats.“ Maybe something like: “People with lower education level did not perceived illegal hunting as a threat for crane survival and is more frequently engaged in this practice.”
Abstract
L23: Please, briefly mention how “severity” was measured.
L24 – L25: Same comment that in lines 11-12
L27 -L30: How is frequency measured? Number of detections per day, month, or year? Sometimes frequency is reported as “average score,” other times as “mean frequency,” or simply “frequency.” Please define the unit of frequency clearly and refer to it consistently. The same with severity.
Introduction
L57 -60: I would suggest deleting these sentences to maintain the narrative flow.
L76: “The decline in migratory crane populations has been alarming,” This sentence refers to a global trend or a Pakistan trend?
L79 – 80: Don’t use capital letters for “Trade” and “Domestication”, “Predation”, and “Collusion.”
L82: In this line and in previous ones, the authors should clarify when they are referring to global trends and when they are referring to country-level trends, as well as distinguish between threats affecting vertebrates in general, birds in particular, and especially cranes.
L87 -93: This final paragraph requires more attention as this state the objective of the study. For example, in the phrase: “…the relative risks posed by these challenges” it is not clear which challenges the authors are referring to. Also, “poaching” already means illegal hunting, so the phrase “known illegal hunting, poaching, and other threats” is redundant. The authors state that they want to evaluate “how poaching and other threats have a significant negative impact on the crane population in Punjab,” but from what is described in the abstract, they did not monitor crane population trends (e.g., mortality, survival, fecundity, etc.) in poaching areas and compare these with equivalent metrics in non-poaching areas to quantify the impact on the population. My understanding is that they evaluated temporal (i.e., frequency) and spatial (i.e., across districts) variation in poaching severity, measured in a way that has not yet been clearly explained. Finally, in the abstract they mentioned social drivers for poaching (i.e. education level) but this aspect is not addressed in the introduction. What other studies on bird poaching or bird illegal trade have found about education level and attitudes towards unsustainable practices like shooting, trapping, or trading birds?
Methods
L115- L116: Please, describe the distribution of responses across stakeholder type. Authors suggest that this information is in Supplementary Material, but the material provided did not contain this information either the identification of the stakeholders involved or the questionary itself.
L122 – L129: As the stakeholders were the main source of information, and the questionnaires the main instrument for collecting it, both aspects need to be described in much greater detail. For example, to which institutions are the “local wildlife officials” affiliated? “Farmers” were present in the same proportion across the districts? Are these all the stakeholders involved in the threats the authors aim to evaluate? The authors did not mention NGOs or community-based organizations. Do these not exist in these districts? Additionally, how were responses distributed across stakeholder types, and what was the gender and age distribution of participants?
In addition, a much more detailed explanation is needed about how the frequency and severity of threats were derived from the questionnaire and how were measured. For example, for frequency and type of threats, did the authors directly ask participants questions such as: “In your opinion, what is the main threat faced by cranes in this area?”, “How frequently did you poach a crane?”, or “How many cranes did you poach in a month, a year?” These questions as well as questions about hunting methods, trafficking regions, and motivations for poaching are sensitive topics. What measures did the authors take to avoid participants concealing data or providing misleading information? Similarly, regarding “severity of threats” how “severity” was measured?
The authors mention that “Additional questions explored hunting practices in relation to cultural traditions and livelihoods.” However, they do not provide any context in the introduction about the traditional use of wildlife in the studied communities, nor do they indicate in the methods section how this sensitive topic was addressed, what type of hunting methods were evaluated, etc. The authors should either include the specific questions used in the manuscript or provide the full questionnaire (in English) as Supplementary Material.
Furthermore, the authors should provide more details about how the questionnaires were implemented and the ethical approval process. Was the selection of participants random, or did it follow a snowball approach? Were the questionnaires self-administered and anonymous, or did an interviewer read the questions and record the answers? Was the questionnaire and its implementation protocol approved by an ethics committee?
L141 – L147: This description of the analysis is far too superficial to allow replication. To begin with, there is no description of data preparation or cleaning, nor of how categorical variables were coded (e.g., threats, hunting methods, and education levels), and distribution of continuous variables (e.g. severity). Second, no explanation is provided for why specific tests or models were chosen, and the description of the models themselves is vague. For example, the authors do not mention which variables were included in each model, which were used as predictors or dependent variables, or whether the variables were continuous, categorical, or binary. There is also no mention of model diagnostics or assumption checking (e.g., normality, homoscedasticity, multicollinearity), or any assessment of model fit.
Author Response

(The authors gave the same response as above.)

Reviewer 3 Report
Comments and Suggestions for Authors
It was a pleasure to review the manuscript “Comparative analysis of threats threshold from various drivers to Cranes along Indus Flyway, Punjab, Pakistan” for biology. This manuscript investigates the intensity and types of threats faced by migratory cranes along the Indus Flyway in Punjab, Pakistan, a lesser studied aspect of population ecology especially when it comes to birds that are distributed in Asia. I think this manuscript has potential to be published in this journal in general pending revisions.
I have provided comments on some major issues in the paper, and I would like to see it one more time after these edits are applied. I did not focus too much on minor comments as the manuscript will probably change once major comments are applied. I think there is room for improvement. After applying these comments, I should be in a better place to leave some minor and fine scale comments. I’ll leave this decision to the editor.
Major Comments:
- After reading the whole manuscript:
- The language in general is good but throughout the manuscript it still can and should be improved. This is an area that could benefit from careful revision. For instance, authors should take responsibility and credit for their work and switch from the passive tone to an active tone.
- Introduction is probably the best part of the paper as it has a story, and it has good cohesion and coherence. There are some minor things I left comments on.
- The term “domestication” appears to be misused throughout the manuscript. Domestication refers to a long-term evolutionary process in which a species undergoes genetic, morphological, and behavioral changes through artificial selection by humans, often over thousands of years or more. True domestication results in heritable traits that distinguish the domesticated population from its wild ancestors. For example, dogs have been fully domesticated from wolves. In contrast, animals that are merely habituated, tamed, or conditioned to human presence such as those raised in captivity or used to human interaction should not be referred to as “domesticated”. I recommend replacing “domesticated” with more accurate term such as “tamed” depending on the context in which it is used.
- Methods and, more specifically, sampling, questionnaire survey and statistical methods should be re-written. There are a lot of details that need to be included. I’ll explain:
- Sampling: Who are the authors referring to when saying stakeholders? Why their opinion matters (I’m not saying it doesn’t matter, but there needs to be some justification), The public? How was the sample size (400) selected? Based on what formula?
- Questionnaire: At its current form it’s not clear what has been measured? How has it been measured? Where were the items taken from? How were they being treated and cleaned? What theory has been used/tested? If theory has been used, then there needs to be these three components: conceptualization (defining), operationalization (measuring), and interpretation (explaining)
- Statistical analysis: Authors only mentioned bunches of different analyses that they used. While more depth is needed on how each analysis has been conducted and how the models fit (e.g. if the data has been tested for correlation, imputation, scaled?). And how was fitness tested? AIC?
- My main concern with this manuscript is that it is not clear what has been tested actually. It seems like residents in those areas were surveyed (not sure how, in person?) about their PERCEPTION of threats frequency and severity that cranes (all species in Pakistan) are facing. However, authors reported that their results like these PERCEIVED threats are the reality (which could be the case, but evidence is going to be needed for such a claim). Also, it is not clear how each of these species in the survey were teased apart and why they all have been treated as the same even though authors mentioned they used pictures to help IDing the species. I think the results and methods should be thoroughly revised and after that I’ll be in a better position to mention other minor comments.
Abstract:
-Line 12: Why illegal hunting and not using the term poaching?
-Line 13: Domestication is a long evolutionary process. I think what you meant here is capturing them and taming? Tokeep them as pets?
-Line 13: Illegal wildlife trade*
-I’ll read on to see how the role of education was assessed.
- Line 14: Not everyone’s gonna know where these places are, so I would suggest using the name followed by a direction (e.g. Northern Pakistan).
-After reading the simple summary I’m still not sure if you studied a single species or multiple species? And what species? Please provide this information.
-I think the abstract jumps into the context of Cranes in Indus Flyway too early. For the sake of having a broader audience authors should start broader and then narrow down to their study species and context.
-Line 24: Not sure what you mean by targeted surveys? Who were targeted?
-Authors should be careful in mentioning their results as they are mere facts. These are the perceived threats according to the population that was surveyed (still unknown who was surveyed).
Introduction
- I like the flow of the introduction. This is what I mean when I mention start broader and then narrow down.
-Line 71: What do mean by graceful? Not a scientific/objective term. Please avoid subjective terms across the manuscript.
- Line 74: Scientific names should be provided.
- Line 89: Again, how do you differentiate between poaching and illegal hunting? I would suggest using the term poaching. These two papers might help:
-Eliason, S. L. (1999). The illegal taking of wildlife: Toward a theoretical understanding of poaching. Human Dimensions of Wildlife, 4(2), 27-39.
- Montgomery, R. A. (2020). Poaching is not one big thing. Trends in Ecology & Evolution, 35(6), 472-475.
Methods:
- Figure 1: I think the figure looks good in general but first inset map needs to be zoomed in and names of the neighboring countries should be shown such as Iran, India etc.
- Figure 1: Site of Visit maybe should be changed to Field Site?
- So still not clear, please make it clear that when you say crane for the rest of the manuscript you are referring to all the species that occur in Pakistan.
- Line 115: Define stakeholders?
- Also it is not clear how the 400 sample was chosen.
- Is there an ethics code? How was consent provided? Ethical code and how consent was being treated should be explained and it is such an important topic in any conservation social science paper. So please take this seriously and explain thoroughly.
- Line 141: R statistical software [40]*.
- Put the citations at the end of the sentences to improve readability.
- It’s not clear what has been analyzed and what was being regressed. Please re-write the statistical analysis section.
- Also not clear to me what the authors mean by frequency and severity of a threat. So you presented the respondents with something like this? How frequent from 1-5 and severe of a threat from 1-5 do you think poaching is?
Results:
-What is figure 2 showing? Positive relationship between severity and frequency? Which makes sense but I’m not sure if violin plot would add anything else, and if not I’m not sure why authors used a violin plot where a simple linear regression would’ve shown the relationship. Am I missing anything?
- It seems like illegal killing/hunting and poaching have been used interchangeably in the draft. Please clarify. Refer to my previous comment.
-Not sure if table 3 is necessary. Put it in the supplementary please.
- Also please put the translated version of the questionnaire in the supplementary.
-There has been no discussion of how common is consuming cranes and their poaching? Please cite this evidence in the introduction. I’m not sure how common of a practice is this in Pakistan. Even if it’s a news article and not a peer-reviewed paper, I think it should be cited as it matters for the context.
- Line 287: Why this model? Why were these variables chosen to be tested? I didn’t see a specific hypothesis. What is the response variable here? Severity (1-5)? Did you make a composite score of the different methods severity by averaging? Please clarify.
Discussion:
I think the main point I have about discussion is better organization. I highly encourage authors to use subheadings to help readers with the points in their discussion. I avoid digging into this part too deeply as it might change drastically after applying the above comments.
Conclusion:
I think authors generalized a lot in their conclusion. Authors should come up with suggestions and recommendations based on their results and avoid mentioning general points such as prioritizing conservation since this is not their finding.

I left them in my general overview above.
Author Response
Dear Reviewer,
Thank you very much for your time and constructive feedback on our manuscript titled "Comparative analysis of threats threshold from various drivers to Cranes along Indus Flyway, Punjab, Pakistan." We greatly appreciate your thoughtful comments, which have significantly contributed to improving the clarity and quality of the manuscript.
We have carefully addressed all the comments point by point. Please find attached the detailed response document, which outlines our responses to each comment, along with the corresponding revisions made in the manuscript. For your convenience, all changes in the manuscript are clearly highlighted by yellow highlighter and the page number and line number is clearly mention in response document. The english Editing and Figures and Tabels settings were majorly considered during revision.
We believe that the revisions have strengthened the manuscript, and we hope that it is now suitable for publication in the journal.
Thank you once again for your valuable suggestions and your continued support.

Reviewer 4 Report
Comments and Suggestions for Authors
The article is valuable in terms of the issues discussed.
The study encompassed a ca. 66 141 km² semi-arid region of Punjab along the Indus and Jhelum rivers, delineating 23 crane distribution sites through systematic field surveys and stakeholder consultations.
Fieldwork, conducted across successive migration cycles between September 2021 and March 2024, employed 400 structured questionnaires administered to wildlife officials, farmers, fishers, utility personnel, hunters, and local residents.
Threat assessments targeted illegal hunting, trafficking, electrocution, and predation, recording crane mortalities from power-line collisions, electrocutions, and jackal or feral dog attacks along with their spatial and temporal context. High-risk zones were identified near feeding and roosting habitats adjacent to medium- and low-voltage distribution lines. This community-based survey approach enabled comprehensive spatial mapping of threats without relying on direct carcass detection.
However, a few clarifications are necessary:
Introduction
The information regarding the human population should be updated, considering that the 8 billion mark was exceeded in 2022. The introduction discusses threats to species; it would be useful to include an explanation of how do the authors conceptualize the term “threat” in relation to the EU Habitats Directive definitions:
Pressure: An activity that is currently having a negative impact on a habitat type or species during the reporting period.
Threat: An activity that is expected or anticipated to have a negative impact on a habitat type or species in the future.
The study refers to the crane species that traverse the Indus Flyway. It is necessary to include the scientific names of these species in the article’s introduction.
Line 78–80: we suggest that the categories of activities impacting crane species be mentioned without capitalization, and that names be checked for correctness—e.g., “Collusion” probably refers to “collision.”
Materials and Methods
We recommend merging Sections 2.2 and 2.3 under the heading Study Design and removing redundant text (specifically the first paragraph on questionnaire deployment). The methodology needs to specify exactly which threat categories were assessed and how—e.g., “severity was evaluated on a scale from 1 to 5, where 1 represents low severity and 5 very high.” It should also explain how respondents’ education levels were evaluated, in alignment with the analyses presented in the Results.
Results
Although the statistical analysis is very valuable, several aspects require additional explanations. In Chapter 3.1, predation—a natural population-regulation mechanism—has been grouped with anthropogenic threats. If stray dogs are included as a consequence of human activity, this should be justified, since the data show low frequency and severity.
For Ch. 3.4, retitle to “Relationship between Mortality and Respondent Type,” and revise Line 234 to read: Grus virgo is the most affected species…
In Figure 3, we suggest to replace “hunting method” with “Threat” or “cause of mortality,” since Jackals and Power Lines are not hunting methods.
Ch. 3.5: It seems strange that Wildlife Officials report a lower education level than Electricity Workers; this discrepancy should be discussed, given that the former group should be more familiar with the study’s subject.
Ch. 3.6: clarify the title as Influence of Education Level on assessing Threat Frequency and Severity
In Chapter 3.8, the relationships between variables are unclear—for example:
“Education positively influences all methods, especially Power Lines (β = 2.13) and Shooting (β = 2.21), suggesting higher education favors alternative, less lethal methods.”
“Predation similarly discourages Live Capturing (β = -26.81), favoring lethal methods.”
Conclusions
Ch. 5 should be expanded to draw out the key conclusions that follow from Ch. 3.
Finaly, this is a valuable study in both thematic focus and methodology, but revisions are required in several areas to fully substantiate and highlight the work’s contributions.
Comments on the Quality of English LanguageWe recommend reviewing and correcting the terms and phrases to convey the authors’ ideas as faithfully as possible.
Author Response
Dear Reviewer,
Thank you very much for your time and constructive feedback on our manuscript titled "Comparative analysis of threats threshold from various drivers to Cranes along Indus Flyway, Punjab, Pakistan." We greatly appreciate your thoughtful comments, which have significantly contributed to improving the clarity and quality of the manuscript.
We have carefully addressed all the comments point by point. Please find attached the detailed response document, which outlines our responses to each comment, along with the corresponding revisions made in the manuscript. For your convenience, all changes in the manuscript are clearly highlighted by yellow highlighter and the page number and line number is clearly mention in response document. The english editing and Figures, tables settings were priorily considered during revisions.
We believe that the revisions have strengthened the manuscript, and we hope that it is now suitable for publication in the journal.
Thank you once again for your valuable suggestions and your continued support.

Round 2
Reviewer 3 Report
Comments and Suggestions for Authors
It was a pleasure to re-review the manuscript “Comparative analysis of threats threshold from various drivers to Cranes along Indus Flyway, Punjab, Pakistan” for biology. Authors did a great job replying politely and professionally to the comments raised in the previous version. There are clear signs of effort to improve the manuscript. The manuscript has certainly improved. However, there are still a couple of major issues remaining. And I think they should be fixed before this manuscript is going to be published. Otherwise, it would be misleading information.
Major Comments:
- Why these two species of cranes were used out of the 4 present species in Pakistan. And how do authors make sure that the respondents are familiar with these species and can distinguish them?
- I noticed that authors have tried to be more consistent in this version, but there are still inconsistencies such as using the term hunting and poaching. It seems like authors are referring to poaching the whole time. Although it’s still not clear to me what hunting practices are? Therefore, I’m not sure what the authors mean when they say there was a “strong association between education and hunting practices”.
- Are there any non-lethal hunting methods? Do you mean capturing methods?
- “shooting and live capture for illegal wildlife trade/tamed”. Should be changed to taming.
- Predation? Do you mean predation by natural predators? If so please clarify.
- The language in general is good and has improved. However, flow changes are abrupt. I’d suggest having a topic sentence and stick to that topic conceptually throughout the whole paragraph. Then try linking it to the next paragraph with a connecting concluding sentence. I know it sounds easier said than done, but that’s an expectation from a good research manuscript.
- Again, the language should be softened when it comes to presenting results. Even in your hypothesis, you said “Poaching is the most sever threat with ….” But the problem is, you won’t be able to verify this hypothesis with your study design and data. Even if all respondents mention poaching is the most severe threat (full agreement 100%), this still would be the perceived severity of a threat not the actual severity of it.
- Line 128: There should be a space between n and numbers: (n = 30).
- Line 124-150: You should report the numbers and median of the age from the sample in the first paragraph of your results where you describe the sample, not in your methods section.
- I repeat this question. If it doesn’t apply there should be a limitation section describing it: Methods and, more specifically, sampling, questionnaire survey and statistical methods should be re-written. There are a lot of details that need to be included. I’ll explain:
- Sampling: Who are the authors referring to when saying stakeholders? Why their opinion matters (I’m not saying it doesn’t matter, but there needs to be some justification), The public? How was the sample size (400) selected? Based on what formula?
- Questionnaire: At its current form it’s not clear what has been measured? How has it been measured? Where were the items taken from? How were they being treated and cleaned? What theory has been used/tested? If theory has been used, then there needs to be these three components: conceptualization (defining), operationalization (measuring), and interpretation (explaining)
- Statistical analysis: Authors only mentioned bunches of different analyses that they used. While more depth is needed on how each analysis has been conducted and how the models fit (e.g. if the data has been tested for correlation, imputation, scaled?). And how was fitness tested? AIC?
- This is from the previous review, but I think it still applies: My main concern with this manuscript is that it is not clear what has been tested actually. It seems like residents in those areas were surveyed (not sure how, in person?) about their PERCEPTION of threats frequency and severity that cranes (all species in Pakistan) are facing. However, authors reported that their results like these PERCEIVED threats are the reality (which could be the case, but evidence is going to be needed for such a claim). Also, it is not clear how each of these species in the survey were teased apart and why they all have been treated as the same even though authors mentioned they used pictures to help IDing the species. I think the results and methods should be thoroughly reviewed and after that I’ll be in a better position to make other minor comments.
Author Response
Dear Reviewer,
I would like to sincerely thank you for your valuable feedback and for highlighting the major concerns in my paper. Your insights have been instrumental in guiding me through the revisions. As a result, I have thoroughly reworked the summary, abstract, introduction, methodology, results, discussion, and conclusion sections to address your comments comprehensively. I hope that the revisions reflect my genuine efforts to improve the quality of the paper. I would be grateful for your kind consideration and hope that the revised manuscript meets your expectations.
Thank you once again for your thoughtful review.

Reviewer 4 Report
Comments and Suggestions for Authors
The revised article successfully reflects the heightened attention to clarity and methodological rigor.
The introduction has been updated with recent data and precise definitions, and the Materials & Methods section now presents a coherent design with detailed descriptions of the threat evaluation criteria and the methods used to quantify respondents’ education levels.
The results are presented logically, with robust interpretations of the statistical relationships. Terminology corrections and the inclusion of scientific names have enhanced the text’s accessibility for the specialist community.
Overall, the work stands out for its consistency, relevance, and substantial contribution to understanding threats to two key migratory crane species, Grus grus and Grus virgo.
Author Response
Dear Reviewer,
Thank you very much for your thoughtful and constructive feedback. We greatly appreciate your positive comments on the clarity, methodological rigor, and structure of the revised article. We are pleased to hear that the updates in the introduction and Materials & Methods section, as well as the improvements in the presentation of results and terminology, have enhanced the manuscript's clarity and accessibility.
Your suggestions have been invaluable in refining the manuscript, and we are confident that the revisions have strengthened the overall contribution of the work to the understanding of threats facing Grus grus and Grus virgo. We are grateful for your time and consideration, and we look forward to the continued success of the publication.